# Evaluation of the Salivary Level of Cortisol in Patients with Oral Lichen Planus: A Meta-Analysis

**DOI:** 10.3390/medicina55050213

**Published:** 2019-05-27

**Authors:** Pia Lopez-Jornet, Elisa Zavattaro, Hamid Reza Mozaffari, Mazaher Ramezani, Masoud Sadeghi

**Affiliations:** 1Facultad de Medicina y Odontologia Universidad de Murcia, Hospital Morales Meseguer, Clinica Odontologic Adv Marques Velez s/n, 30008 Murcia, Spain; majornet@um.es; 2Dermatology Unit, Department of Translational Medicine, University of Eastern Piedmont “Amedeo Avogadro”, 28100 Novara, Italy; elisa.zavattaro@med.uniupo.it; 3Department of Oral and Maxillofacial Medicine, School of Dentistry, Kermanshah University of Medical Sciences, Kermanshah 6713954658, Iran; mozaffari20@yahoo.com; 4Medical Biology Research Center, Kermanshah University of Medical Sciences, Kermanshah 6714415185, Iran; 5Molecular Pathology Research Center, Imam Reza Hospital, Kermanshah University of Medical Sciences, Kermanshah 6714415153, Iran; mazaher_ramezani@yahoo.com; 6Students Research Committee, Kermanshah University of Medical Sciences, Kermanshah 6715847141, Iran

**Keywords:** oral lichen planus, saliva, cortisol, meta-analysis

## Abstract

*Background and objective*: Cortisol, as the main human glucocorticoid, is considered to be a biological marker of stress and anxiety. Since it is known that oral lichen planus (OLP) can appear and worsen during stressful events, cortisol levels have been previously studied in OLP patients. The present meta-analysis aims to assess the salivary concentration of cortisol in OLP patients compared to healthy controls. *Materials and methods*: Web of Science, PubMed, Cochrane Library, and Scopus databases were searched up to October 2018. The RevMan 5.3 software was used with the mean difference (MD) and 95% confidence intervals (CIs). The CMA 2.0 Software was used to evaluate the publication bias, sensitivity analysis, and meta-regression as possible sources of heterogeneity. *Results:* 10 studies were analyzed and a total of 269 OLP patients and 268 controls were included. The pooled MD of the salivary levels of cortisol in OLP patients compared with controls was 4.27 ng/mL (95% CI: 2.33, 6.21; *P* < 0.0001), thus, the salivary level of cortisol in OLP patients was significantly higher than in controls. In Indian-based population studies, a significant difference in the salivary cortisol levels in OLP patients compared with controls was detected (MD = 5.62 ng/mL; 95% CI: 2.67, 8.56; *P* = 0.0002). In addition, a significant difference in the salivary cortisol levels in the OLP patients compared with the controls was obtained in studies performed with enzyme-linked immunosorbent assay (ELISA) method (MD = 5.33 ng/mL; 95% CI: 2.72, 7.93; *P* < 0.0001). *Conclusion:* We suggest that supportive psychological treatment together with the conventional therapy could increase patients’ capability to prevent stress, anxiety, and depression.

## 1. Introduction

Cortisol is the main glucocorticoid in humans and has a wide range of effects on metabolism, vascular responsiveness, immunoregulation, cognition, and behavior [1]. Stress, as a psychological factor, causes an elevation in the levels of cortisol, and this alteration is associated with an imbalance of cytokines, which can predispose to the development of autoimmune diseases [2,3]. Oral lichen planus (OLP) is a chronic, inflammatory disease in which cytokines can play a role in its progression and etiology [4,5]. The OLP incidence is higher in females than in males and has a different age range around the world [6]. This disease is correlated with salivary oxidative stress [7]. Cortisol has been considered as a biological marker of stress and anxiety, and its salivary levels have been studied in relation to psychological changes in OLP patients, but with contradictory results [8]. The anxiety is another psychological factor that can increase the cortisol levels [9]. It is recognized that stressful events occur before the onset of OLP in about 10–68% of cases. Additionally, the severity of symptoms may be worse or increased during stress [10,11,12]. The aim of this meta-analysis study is to evaluate the salivary cortisol levels in the OLP patients compared to the controls.

## 2. Material and Methods

### 2.1. Protocol and Search Strategies

This study was approved according to the Preferred Reporting Items for Systematic Reviews and Meta-Analyses (PRISMA) guidelines [13]. The databases of Web of Science, PubMed/Medline, Scopus, and Cochrane Library were comprehensively searched and other databases manually with key terms (“OLP” OR “oral lichen planus”) AND (“cortisol”) AND (“salivary” OR “saliva”) up to October 2018, without any language restriction.

### 2.2. Study Selection

Two reviewers (M.S and P.L.J) contributed to the study selection. The first reviewer evaluated the studies to check if they followed the inclusion criteria. The second reviewer re-evaluated. The inclusion criteria were: (i) Detection of cortisol levels in the saliva (fasting state and before diurnal changes) of the OLP patients with/without LP and healthy control groups (case-control or comparative cross-sectional studies); (ii) the OLP diagnosis was in accordance with the clinical and/or histopathological World Health Organization (WHO) criteria [14], (iii) the healthy controls were reported without other skin and/or systemic diseases affecting the OLP patients.

### 2.3. Data Extraction

One reviewer (M.S) extracted the relevant data of every study. The second reviewer (E.Z) re-checked them. The extracted data for all studies included in the meta-analysis are shown in Table 1.

### 2.4. Quality Evaluation

One author (M.R) estimated the quality of each involved study using the Newcastle–Ottawa Quality Assessment Scale (NOS), with score ≥7 being high quality [15].

### 2.5. Statistical Analyses

The Review Manager 5.3 (RevMan 5.3, The Cochrane Collaboration, Oxford, United Kingdom) applying the mean difference (MD) and 95% confidence intervals (CIs) was used with a random-effects analysis. The pooled MD of the studies was calculated to estimate the salivary cortisol levels of the OLP patients versus the controls. The Q and I^2^ statistics were applied to evaluate heterogeneity between the studies. With regard to the Q statistic, heterogeneity was determined if *P* < 0.1 (or I^2^ > 50%). *P*-value (two-tailed) < 0.05 was considered statistically significant. The Comprehensive Meta-Analysis Software version 2.0 (CMA 2.0) was used to evaluate the publication bias among the studies by funnel plot by Begg’s and Egger’s tests, sensitivity analysis, and meta-regression as possible sources of heterogeneity that *P* < 0.05 showed a significant existence of publication bias. The unit of measurement of cortisol was ng/mL in saliva. The *Z*-test was used to show the significance of the pooled MD. Two analyses, namely removing one study and cumulative analysis, were used for sensitivity analysis to figure out the results stability. The meta-regression was done with the *P*-value and regression coefficient (*r*) to assess the strength of the association between the study period and the pooled MD of the salivary cortisol levels. The mean (range) and median (quartile) were estimated to mean ± SD [16,17].

## 3. Results

The selection of the studies was done based on Figure 1. Out of 45 studies identified in the databases, 10 studies were selected and entered in the meta-analysis.

Characteristics of 10 studies involved in the meta-analysis are presented in Table 1. The studies were published from 2003 to 2017 and included 269 OLP patients and 268 controls. Four studies were reported from India [1,18,19,20], three from the Middle East [21,22,23], two from Europe [24,25], and one from Brazil [8]. In six studies [1,18,21,22,23,24], the detection method of salivary cortisol level was enzyme-linked immunosorbent assay (ELISA), whereas the remaining methods were radioimmunoassay in one study [8] and the chemiluminescence immunoassay in three studies [19,20,25]. Eight studies [1,8,18,19,21,22,23,24] had case-control design and two studies [20,25] had comparative cross-sectional design. Other data of the studies are shown in Table 1. According to the NOS criteria, eight studies showed high quality. Three studies [1,8,23] reported that few OLP patients had LP, one study [22] excluded LP patients, and other studies did not report this problem.

### 3.1. Meta-Analysis

#### 3.1.1. Salivary Cortisol

The pooled MD of the salivary cortisol levels in OLP patients compared with healthy controls was 4.27 ng/mL (95% CI: 2.33, 6.21; *P* < 0.0001) with I^2^ = 96% (*P*_h_ < 0.00001) (Figure 2). The level of salivary cortisol in OLP patients was significantly higher than in controls.

#### 3.1.2. Subgroup Analysis

The subgroup analysis was done based on two terms: geographical area of participants and detection method of salivary cortisol (Table 2). There was a significant difference only in salivary cortisol levels between OLP patients and controls in India (MD = 5.62 ng/mL; 95% CI: 2.67, 8.56; *P* = 0.0002). There was a significant difference in the salivary cortisol levels in OLP patients compared with controls performed with ELISA method (MD = 5.33 ng/mL; 95% CI: 2.72, 7.93; *P* < 0.0001) and not for studies reported from Middle East and Europe or with other detection methods.

### 3.2. Publication Bias

The funnel plot of the overall analysis is presented in Figure 3. The tests didn’t confirm any publication bias across the studies. The *P*-values were 0.654 and 0.886 for Begg’s and Egger’s tests, respectively.

### 3.3. Sensitivity Analysis

The sensitivity analyses by removing one study and cumulative analysis were done on overall analysis. They revealed that the results did not change, and thus the acquired results were stable.

### 3.4. Meta-Regression 

Meta-regression analysis showed a significant statistical correlation between the pooled MD of the salivary cortisol levels versus the year of publication (Figure 4). The result showed that the year of publication was not one of the reasons for heterogeneity (Correlation coefficient (*r*) = 0.046, *P*-value = 0.598).

## 4. Discussion

Interaction of genetic and environmental factors, study patients’ lifestyle, and stress can affect the OLP etiopathogenesis [8,18,26,27], as well as anxiety [8,26,27]. The present meta-analysis demonstrated that salivary cortisol level in patients affected by OLP was significantly higher than healthy controls and, when considering the geographical origins of patients, such difference was detected only in studies involving the Indian population. Moreover, the samples processed by the ELISA method showed higher salivary cortisol levels in OLP patients compared with controls, and the difference was significant. In this regard, lack of significance with the other used methods may be due to the low number of reported studies. Out of 10 studies reported [1,8,18,19,20,21,22,23,24,25] in the present meta-analysis, six studies [1,18,19,21,23,24] showed a higher significant level of salivary cortisol in the OLP patients compared with the controls. 

Concerning different clinical OLP forms, one study [10] reported higher serum cortisol levels in patients affected by the erosive variant of OLP, however there was no difference in patients with the reticular form compared with the controls. Lopez-Jornet et al. [25] did not find any significant difference in salivary cortisol levels between different OLP clinical variants (reticular-papular and atrophic-erosive), unless this can be due to fewer patients with erosive OLP. It can be said that reticular and papular lesions are asymptomatic and that the patient doesn’t feel burning and/or pain. Therefore, the serum cortisol and subsequently the salivary cortisol levels remain in the normal range [22]. 

Cortisol is a biomarker of stress and anxiety, hence its salivary level and its relationship with psychiatric disorders in the OLP patients have been studied, even with controversial results [8,18,21,28]. Koray et al. [21] showed higher levels of salivary cortisol and anxiety before biopsy in the OLP patients. This situation has probably increased the levels of studied variables, and Nosratzehi et al. confirmed this data as well as stress and serum cortisol level [22]. Opposite to this, Rodstrom et al. did not find any correlation between cortisol and stress levels [28]. The evaluation of the salivary cortisol levels and anxiety reflecting stress seems to be a good parameter in OLP research [21]. Accordingly, OLP can worsen at times of mental stress [29]. Shah et al. showed a correlation between high levels of cortisol in OLP patients and depression, anxiety, and stress, and such result was in line with those of previous studies [21,30]. Therefore, a positive relationship between salivary cortisol and anxiety levels in OLP patients shows that psychological treatment combined with traditional treatment may be useful in decreasing OLP severity [18]. On the other side, some studies failed to detect a correlation between stress intensity, anxiety scores and salivary cortisol levels in OLP patients [23,31]. The used method and ethnicity were two effective significant factors on the salivary cortisol levels. 

Additionally, it can be argued that higher salivary cortisol levels were observed in patients compared with healthy controls. In fact, because of their immunosuppressive effect, cortisol and its derivatives are currently used for topical administration in the treatment of OLP [32]. With this in mind, it can be expected that higher levels could determine OLP healing or mild forms of the disease. In addition, one study in a general population reported that salivary cortisol in women was higher than men, and older individuals had more salivary cortisol in both genders [33]. Another study [34] showed that there are several strategies for salivary cortisol collection and, among these strategies, time of sampling can be one of important factors. Unfortunately, none of the studies included in the meta-analysis have reported the association of these factors with the salivary levels of oral lichen planus. Therefore, the future studies need to check these associations in OLP patients.

In the present meta-analysis, there were two important limitations, namely the difference in the used methods and the different prevalence of OLP forms in the considered studies.

## 5. Conclusions

With regard to higher salivary levels of cortisol in OLP patients compared to controls, we could state that supportive psychological treatment together with the conventional therapy could increase patients’ capability to prevent stress, anxiety, and depression and promote OLP healing. To demonstrate this hypothesis, further researches are needed with bigger cohorts in order to evaluate all the different variants of OLP, especially erosive form. In this meantime, ethnicity should also be considered.

## Figures and Tables

**Figure 1 medicina-55-00213-f001:**
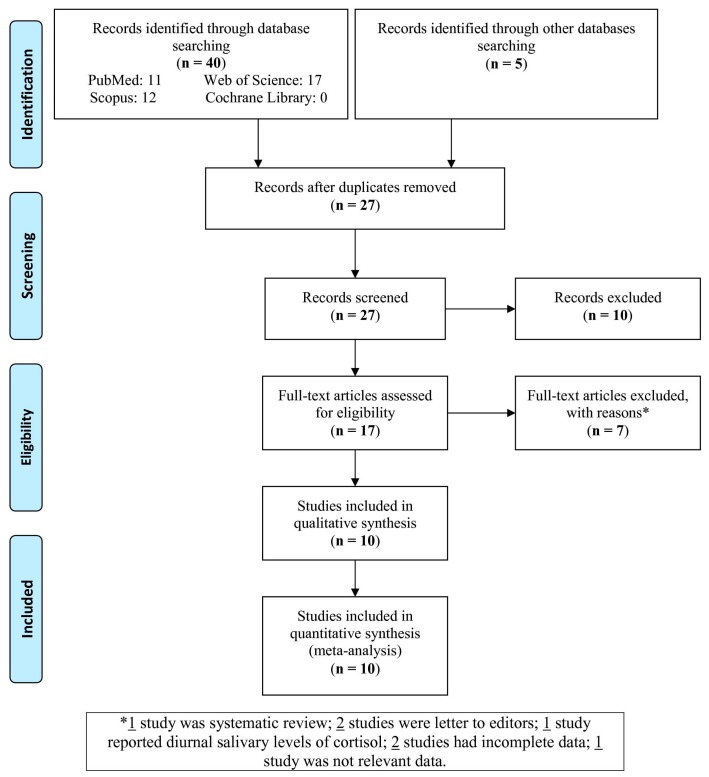
Flowchart of the study selection.

**Figure 2 medicina-55-00213-f002:**
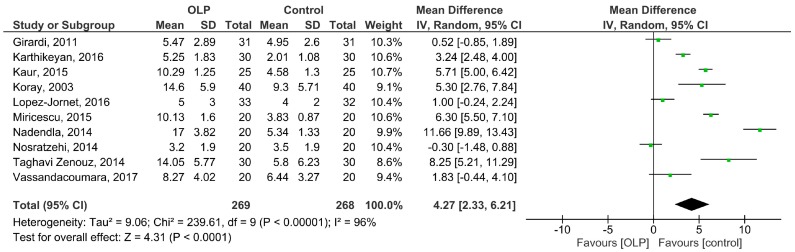
Forest plot of random-effects analysis: Salivary cortisol levels in the OLP patients vs. the controls. The green box on each line shows the mean difference (MD) for each study and the black diamond at the bottom of the graph shows the overall MD of the ten studies. **Abbreviations**: OLP, oral lichen planus; SD, standard deviation; CI, confidence interval.

**Figure 3 medicina-55-00213-f003:**
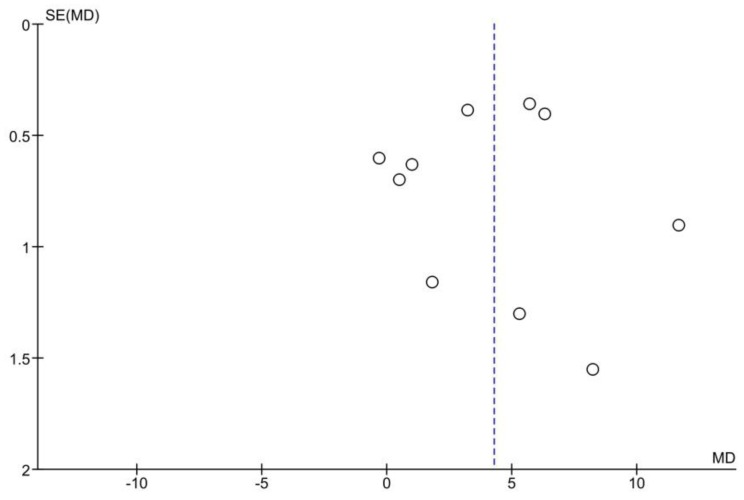
Funnel plot of random effect analysis: Salivary cortisol levels in the OLP patients vs. the controls. Abbreviations: MD, mean difference; SE, standard error

**Figure 4 medicina-55-00213-f004:**
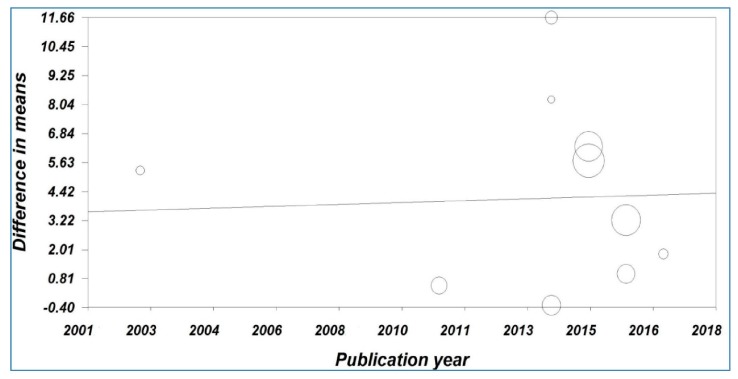
Meta-regression analysis of the study period versus the mean difference of the salivary cortisol levels.

**Table 1 medicina-55-00213-t001:** Features of the studies involved in the meta-analysis (*n* = 10).

The First Author, Year	Country	Study Design	No. of OLP Patients/Controls	Mean Age of OLP Patients/Controls, Year	Male% of OLP Patients/Controls	Measurement Method (System)	Sampling Time (am)	OLP Forms	Score *
Koray, 2003 [21]	Turkey	CC	40/40	35.2/34.6	40/45	ELISA (Diagnostic System Laboratories, Inc, Webster, TX, USA)	9–9:15	NA	7
Girardi, 2011 [8]	Brazil	CC	31/31	53.8/55.5	13/13	Radioimmunoassay (Siemens Medical Solutions Diagnostics, Los Angeles, California, USA)	8–10	55% combination of reticular and atrophic erosive, 25.8% reticular, 9.6% atrophic-erosive, and 9.6% combination of papular, reticular, and plaque	7
Nadendla, 2014 [18]	India	CC	20/20	Matched/Matched	Matched/Matched	ELISA (Diametra kit, Korea)	9–9:15	NA	8
Nosratzehi, 2014 [22]	Iran	CC	20/20	45.8/42.8	35/Matched	ELISA	9–10	NA	8
Taghavi Zenouz, 2014 [23]	Iran	CC	30/30	48.6/47.3	67/Matched	ELISA (DRG Salivary Cortisol—HS ELISA SLV 4635 DRG Instruments, GmbH, Germany) using Hyperion (USA)	9–10	NA	7
Kaur, 2015 [1]	India	CC	25/25	Matched/Matched	36/Matched	ELISA	9–10	52% reticular, 44% erosive, and 4% plaque	7
Miricescu, 2015 [24]	Romania	CC	20/20	NA/NA	75/NA	ELISA	9–10	100% keratosis and atrophic/erosive	6
Karthikeyan, 2016 [19]	India	CC	30/30	39.9/NA	43.3/50	Electro chemiluminescence immunoassay (ECLIA)	before 10	76.7% reticular, 13.3% erosive, and 10% linear	6
Lopez-Jornet, 2016 [25]	Spain	CCS	33/32	57/53	21.2/25	Chemiluminescent enzyme immunoassay (Immulite; Siemens, Erlangen, Germany)	10–12	75.7% reticular-papular and 24.3% atrophic-erosive	7
Vassandacoumara, 2017 [20]	India	CCS	20/20	42.3/34.1	65/50	Chemiluminescent immunoassay (ADVIA^®^ Centaur™ System)	8–9	NA	8

**Abbreviations**: ELISA, enzyme-linked immunosorbent assay; OLP, oral lichen planus; NA, not available; CC, case-control; CCS, comparative cross-sectional. * Newcastle–Ottawa Quality Assessment Scale (NOS) score.

**Table 2 medicina-55-00213-t002:** Random-effects analysis of salivary cortisol levels in oral lichen planus patients vs. the controls based on area of participants and measurement method of cortisol.

		No. of Studies	MD (95% CI)	*P*-Value	I^2^	P_h_
Area	India	4	5.62 (2.67, 8.56)	**0.0002**	97%	<0.00001
	Middle East	3	4.28 (−1.13, 9.70)	0.12	95%	<0.00001
	Europe	2	3.67 (−1.52, 8.87)	0.17	98%	<0.00001
Method	ELISA	6	5.33 (2.72, 7.93)	**<0.0001**	97%	<0.00001
	Others	3	1.93 (−0.01, 3.87)	0.05	83%	0.002

**Abbreviations**: MD, mean difference; ELISA, enzyme-linked immunosorbent assay; CI, confidence interval; P_h_, P_heterogeneity_.

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
