# Peer review of "Evaluation of the Salivary Level of Cortisol in Patients with Oral Lichen Planus: A Meta-Analysis"

_medicina, 2019, doi:10.3390/medicina55050213_

Round 1

Reviewer 1 Report

An interesting review article in which the authors performed a meta-analysis of the evaluation of salivary level of cortisol in patients with oral lichen planus.

Following are my comments for this article if they can be addressed:

Page 2, line 45 …………………..The OLP incidence is higher in females than in males and has a different age range around the world.

Why was the male % of OLP patients taken into consideration in Table 1? Would the salivary cortisol levels be different in males than in females?

Did the sampling time of the day have any effect on salivary cortisol levels in patients with oral lichen planus?

Did any of the patients with oral lesions have skin lesions of lichen planus?

Page 3, line 98…………………….”In six studies the measure method of salivary cortisol was ELISA……………immunoassay.”

Table 2 – change no. of studies with ELISA to 6, instead of 7.

4. Moderate grammatical concerns/examples of awkward sentence structure are noted. A thorough review for content, composition and grammar is recommended.

Author Response

Dear Reviewer

Thanks for your valuable comments:

Page 2, line 45 …………………..The OLP incidence is higher in females than in males and has a different age range around the world.

 Why was the male % of OLP patients taken into consideration in Table 1?

 Re: In Table 1 we have reported the male % of OLP patients in the evaluated studies (the female % could be calculated). According to the OLP higher incidence in females, the percentage of males in the considered studies resulted lower than 50% in 6 studies up to 10.

Would the salivary cortisol levels be different in males than in females?

Re: In addition, one study in a general population reported that salivary cortisol in women was higher than men and also older individuals had more salivary cortisol in both genders [33]. But, the studies included in the meta-analysis did not explain the effect of age and sex on the concentration of salivary cortisol in OLP patients.

Did the sampling time of the day have any effect on salivary cortisol levels in patients with oral lichen planus?

 Re:  Another study [34] showed that there are several strategies for salivary cortisol collection and among these

strategies; time of sampling can be one of important factors. But, the studies included in the meta-analysis did  not explain the effect of sampling time on the concentration of salivary cortisol in OLP patients.

Did any of the patients with oral lesions have skin lesions of lichen planus?

Re: Three studies [1,8,23] reported that few OLP patients had LP, one study [22] excluded LP patients, and other studies did not report this problem. 

Page 3, line 98…………………….”In six studies the measure method of salivary cortisol was ELISA……………immunoassay.”

Table 2 – change no. of studies with ELISA to 6, instead of 7.

Re: The change has been done accordingly.

4. Moderate grammatical concerns/examples of awkward sentence structure are noted. A thorough review for content, composition and grammar is recommended.

Re: The paper has been carefully revised and the changes are reported in the revised version.

Reviewer 2 Report

Major points

This is a well surveyed review article of salivary cortisol in OLP and health peoples. Unfortunately, I cannot read a peculiar argumentation of authors compared with other review articles.

     There are many papers regarding cortisol levels in OLP but it seems that we have not been reached to consensus of its clinical relevance. It is reasonable that salivary cortisol increases in level because OLP, especially erosive OLP, which is a severe type of OLP.  Therefore, the intraoral manifestation of OLP could give anxiety or stress to patients, which might result in increasing salivary cortisol. However, the clinical importance of cortisol assessment does not arise from this paper. Regarding your conclusion, do you mean that the measurement can be used for assessment of stress or disease condition in order to give psychological treatment?  

I recommend that authors clearly state your opinion on a cortisol assessment and its clinical relevant. 

Minor points

On lines 19 and 47: Please change “considered a biological marker” to “considered as or to be a biological marker”.On line 44: The description as “OLP is a chronic, autoimmune inflammatory disease” cannot not be fully accepted because all OLP have not autoimmune backgrounds.

On lines 83 and 87: Please make “P” italic.

On lines 184 and 185: The meaning of this sentence is obscure. Do you mean that OLPs with higher cortisol levels show good prognosis, don't you?

Author Response

Dear Reviewer

Thanks for your valuable comments.

I recommend that authors clearly state your opinion on a cortisol assessment and its clinical relevant. 

 Re: The present article is a meta-analysis: it means  that the aim is not to state the author’s opinion, but to examinate the data from multiple independent studies in order to determine overall trends. In our paper we examined 10 different studies and, after appropriate statistical analysis, we obtained that the cortisol salivary concentration was significantly higher in patients compared to healthy controls. When examining different studies on the basis of the geographical area and of detection method, a significant higher concentration of salivary cortisol resulted in studies performed in Indian population and with ELISA method.

In the literature, some studies have assessed the relationship between OLP variants and cortisol level (either in serum or in saliva) and between cortisol level and anxiety, but with controversial results. Additionally, it seems very surprising that higher cortisol level was detected in patients with OLP (and in some cases, in subjects affected by more severe forms of OLP – namely the erosive variant-). In fect, as discussed in the Conclusion/Discussion, cortisol and its derivatives are currently used as effective treatment in OLP. With this in mind, one can expect that high cortisol concentration could be of help improving and/healing such disease. This has been properly discussed in our paper.

In conclusion, we have demonstrated that patients affected by OLP present higher concentration of salivary cortisol compared to healthy controls, but we cannot state that cortisol level coud be of clinical importance, neither in evaluating the OLP severity, nor to allow he measurement of stress. We are aware that further studies are needed in such field.

Minor points

On lines 19 and 47: Please change “considered a biological marker” to “considered as or to be a biological marker”.

Re: Those changes have been edited accordingly

On line 44: The description as “OLP is a chronic, autoimmune inflammatory disease” cannot not be fully accepted because all OLP have not autoimmune backgrounds.

Re: To date, the cause of OLP has not fully understood, and both genetics and immunity may play a role. It is suggested that a reaction of the surface of skin  and/or mucosa to an unknown antigen could be involved.

See also the following refs: Lavanya N et al. Oral lichen planus: an update on pathogenesis and treatment. J Oral Maxillofacial Pathol 2011;15(2):127-132.

Cassol-Spanemberg J et al. Oral lichen planus and its relationship with systemic diseases. A review of evidence. J Clin Exp Dent 2018;10(9):e938—e944.

For such reason, we have reported the above mentioned sentence. Neverthless, in the revised version of the paper, the word “auoimmune” has been deleted.

On lines 83 and 87: Please make “P” italic.

Re: The changes has been edited accordingly

On lines 184 and 185: The meaning of this sentence is obscure. Do you mean that OLPs with higher cortisol levels show good prognosis, don't you?

Re: We have discussed that, since cortisol and its derivatives are currently effective in the OLP treatment (either for topical use), we could expect that high concentrations of salivary cortisol could promote healing and/or improvement of clinical disease. This is only a mere hypothesis, but we are not able to state it.

Reviewer 3 Report

The main concerns of this meta-analysis are listed as following:

What were the methods, what resources were used, what search terms, how many references were found, how many references were actually included in the review, did the review follow PRIZMA guidelines, does the journal require PRIZMA guidelines be followed?  These points should be addressed.

Different study design has different level of evidence. Please list level of evidence for every paper.

Cohort and case-control study are different study design. You must purify them, then collect same level papers to analyze is advised.

Author Response

Dear Reviewer

Thanks for your valuable comments:

The main concerns of this meta-analysis are listed as following:

What were the methods, what resources were used, what search terms, how many references were found, how many references were actually included in the review, did the review follow PRIZMA guidelines, does the journal require PRIZMA guidelines be followed?  These points should be addressed.

Re: Methods are located in Table 1. Web of Science, PubMed, Cochrane Library, and Scopus databases wew used. Key terms (“OLP” OR “oral lichen planus”) AND (“cortisol”) AND (“salivary” OR “saliva”)  were used. Out of 45 references identified in the databases, of which ten references were actually included in the meta-analysis. The article follows PRISMA guidelines. The journal needs PRISMA guidelines. All points were addressed in text (methods and results sections and also Figure 1).

Different study design has different level of evidence. Please list level of evidence for every paper.

Re: All evaluated studies included case and control groups (case-control or comparative cross sectional studies). In order to better explain such concept, we added new column to Table 1.

Cohort and case-control study are different study design. You must purify them, then collect same level papers to analyze is advised.

Re: There was no cohort study (see the answer to the comment above)